# Ultrasound Technology as Inactivation Method for Foodborne Pathogens: A Review

**DOI:** 10.3390/foods12061212

**Published:** 2023-03-13

**Authors:** Carlotta Lauteri, Gianluigi Ferri, Andrea Piccinini, Luca Pennisi, Alberto Vergara

**Affiliations:** Section of Food Inspection, Department of Veterinary Medicine, School of Specialization in Inspection of Foods of Animal Origin, “G. Tiecco” University of Teramo, 64100 Teramo, Italy

**Keywords:** ultrasound, hurdle technologies, foodborne pathogens, non-thermal technology, public health

## Abstract

An efficient microbiological decontamination protocol is required to guarantee safe food products for the final consumer to avoid foodborne illnesses. Ultrasound and non-thermal technology combinations represent innovative methods adopted by the food industry for food preservation and safety. Ultrasound power is commonly used with a frequency between 20 and 100 kHz to obtain an “*exploit cavitation effect*”. Microbial inactivation via ultrasound derives from cell wall damage, the oxidation of intracellular amino acids and DNA changing material. As an inactivation method, it is evaluated alone and combined with other non-thermal technologies. The evidence shows that ultrasound is an important green technology that has a good decontamination effect and can improve the shelf-life of products. This review aims to describe the applicability of ultrasound in the food industry focusing on microbiological decontamination, reducing bacterial alterations caused by food spoilage strains and relative foodborne intoxication/infection.

## 1. Introduction

The European Food Safety Authority (EFSA) reported 5175 foodborne outbreaks from 2015 to 2019 [1]; the Centers for Disease Control and Prevention (CDC) publishes yearly reports that highlight interesting data: 48 million people become ill due to foodborne diseases (128,000 are hospitalized, with 3000 deaths [2]). Due to the increase in outbreak numbers, it is necessary to develop efficient food chain surveillance and adequate microbiological decontamination protocols to guarantee safe food products for consumers. To achieve safety and genuineness, food processing technologies represent essential tools for microbiological control and products’ shelf-life enhancement [3,4]. Due to consumers’ growing requests for “*minimally processed products*”, the food industry applies new technologies to produce safe food matrices that maintain “*fresh-like*” characteristics [5].

Indeed, in conventional technologies, such as thermal treatments, this concept is not applicable: pasteurization and sterilization, commonly used in food industries, cause color alterations, characteristic flavors and a decrease in nutritional value [6,7,8].

Therefore, the food industry and scientific researchers have evaluated alternative non-thermal technologies (NTTs) that maintain the aroma, nutrient value, texture and color while decreasing bacteria that cause spoilage. Tiwari and coworkers defined NTTs as procedures, performed at efficient sublethal or ambient temperatures, that lead to minimal or no impacts on nutritional and quality food parameters [9] (see Figure 1).

The aim of this review article is to perform an analysis of recent discoveries concerning ultrasound technology application in the food matrix’s shelf-life prolongation (bacterial load decrease) and bacterial foodborne pathogen inactivation, and to demonstrate its applicability as a useful green food technology among physical devices.

## 2. Ultrasound: Mechanisms of Action Applied in Food Industry

Ultrasound is a form of vibrational energy produced by a transducer converting electrical energy into acoustic energy. It is a wave that exceeds the human hearing threshold [10]. Basing on the frequency, ultrasound can be classified as follows: power ultrasound (20–100 kHz), high-frequency ultrasound (100 kHz–1 MHz), and diagnostic ultrasound (1–500 MHz) [11]. At medium frequencies (200–500 kHz), chemical effects are prevalent, and collapse is less violent. On the other hand, at high frequencies (>1 MHz), chemical and physical effects decrease and cavitation is minimal; in this case, acoustic flow is predominant [11,12]. In the food industry, power ultrasound is commonly used with a frequency between 20 and 100 kHz to obtain an “exploit cavitation effect” [13]. The molecules are compressed and rarefactive when ultrasound is spread through any medium. Alternative pressure changes cause bubble formation in a liquid medium. There are physical and chemical effects correlated with ultrasound: agitation, vibration, pressure, shock waves, shear forces, microjets, compression and rarefaction, acoustic streaming, cavitation, and the formation of free radicals [14].

This phenomenon of the creation of small vapor bubbles (cavities), expansion, and implosive collapse in ultrasonically irradiated liquids is named “acoustic cavitation” [13,15,16]. There are two types of bubbles: transient and stable [17]. Under ultrasound action, bubbles oscillate, grow, and collapse asymmetrically, forming microjets. Outburst produces pressure shocks up to several 1000 atm, strong shock waves with 400 km/h microjets, and the production of hot spots with a 5000 K temperature; the mechanical effects predominate over the chemical ones [18,19]. In the reaction environment, three different phases have been identified: inside the bubble cavity gas environment, the liquid–bubble interface, and the liquid. In the first phase, there are pyrolysis reactions. In the second and third ones, radicals can occur. In the aqueous environment, the most frequently encountered phenomenon is the formation of the hydroxide radical OH-. It is highly reactive and attacks organic substrates or OH- and recombines with another OH- radical, forming H_2_O_2_. In the interphase area, the temperature is very high; therefore, the occurring reactions are thermal degradation and solute reactions with OH- radicals. Small bubbles are generated by the diffusion of these radicals due to the cavitation bubble’s disruption. In the interphase zone or in liquid nonvolatile solutes, reactive and volatile solids penetrate the bubble and degrade during collapse [19]. Physical and chemical effects are the basis for ultrasound’s application in the food industry [12].

New technologies, such as as vacuum cooling technology, high-pressure processing, ultrasound, and pulsed electric field technology, could guarantee safe and high-quality products. The aim of these new technologies is to reduce processing times, save energy and solvents, and improve the products’ shelf-life. From a “green” methodological point of view, ultrasound-assisted extraction has huge potential as an emergent and innovative technology. It has a low environmental impact, due to decreasing CO_2_ emissions, reducing time, and not presenting toxic effects towards human health [20]. There are two types of ultrasound systems applied in the food industry: contact and non-contact [21]. The first one employs liquids as transmission media and generates waves that have chemical and physical effects in food matrices. This technology is employed for different activities: the extraction of bioactive substances [22], the enhancement of drying rates [23] and freezing rates [24], the degassing of liquids [25], fat separation [26], power hydration [27], the intensification of heat and mass transfer [28], emulsification [29], and liquid food pasteurization [30,31]. Nevertheless, when using this technology, erosion could produce effects on the radiating surfaces and cause the consequent contamination of sonicated food [32]. However, new inert materials such as quartz, Pyrex, ceramics, and polyether could limit the use of metal horns, which are instruments used to evaluate ultrasonic irradiation in different materials (see Table 1).

Wang and coworkers [39] underlined that US has a positive effect in decreasing frying times, in the improvement of cooking yields, and improving the sensory evaluation of meat. The major consequences of ultrasound’s irradiation within a liquid are cavitation and agitation. These two factors are useful in improving heat transfer and freezing rates and accelerating freezing processes [54,55]. In the last mentioned process, there are primary and secondary nucleations: the first allows crystal formation in a solution where crystals are not detected. Primary nucleation can take place in two categories: homogenous and heterogeneous. Homogeneous nucleation occurs when the nuclei are formed spontaneously from the random density fluctuation. On the other hand, heterogeneous nucleation occurs due to the presence of solid impurities that form stable surfaces for nuclei formation, and secondary nucleation takes place where pre-existing crystals are present [56]. Ultrasonic application improves drying in all food matrices [56,57].

There are many advantages: water is removed easily, improving water diffusion from the interior to the product surface; intracellular and extracellular cavitation provides new microchannels; US creates air turbulence to remove moisture; it accelerates the process without a temperature increase [57]. This technology can be employed as a pre-treatment: in fact, many authors underlined that US pre-treatment improved the drying period [58]. As previously mentioned, waves involve a rate mass transfer by physically breaking down tissues and the formation of microchannels [20]. Ozuna and coworkers [59] evaluated the improvement in solute distribution during marination, and changes in water retention capability. McDonnel et al. [60] also underlined the possibility of conserving food sensory properties through these methods.

Iguglia et al. [61] investigated how different US frequencies can influence chicken marination times in terms of meat quality, texture, and lipid oxidation.

The applicability of US in seafood products has been evaluated: Pedròs- Garrido et al. [62] investigated US usage (30 kHz for 5 to 45 min) in different fish (salmon, mackerel, cod, hake). They noticed a major reduction in microbiological spoilage in oily fish, due to having higher fat content, which impacted bacterial decontamination. After 45 **min** of US treatment, there was a reduction in thiobarbituric acid reactive substances; on the other hand, lipids did not show changes.

US has been used for the tenderization of fish: Chang and Wang [53] found that US application for 60 to 90 min in cobia (Rachycentron canadum) improved the time required for tenderizing compared with the traditional aging process and optimized the firmness.

Non-contact technology, known also as the “*air-couple technique*”, uses a medium to ensure a gap between the transducer and the foodstuff. However, there are some drawbacks, such as the mismatch of the acoustic impedance magnitude between air and matrices [63].

## 3. Mechanism of Ultrasound Action against Microorganisms

Thermal treatments are the conventional method to inactivate microorganisms. However, they could lead to reduced sensory quality and nutrient substances [64]. In the last few years, the employment of ultrasound, as a method of decontamination, has been increasing in the food industry to decrease bacterial homeostatic mechanisms. Indeed, if ultrasound is combined with another technology that sensitizes the microorganism structure to the action of ultrasonic waves, microbial disruption, and, consequently, inactivation, will be probably enhanced. On the other hand, ultrasound application will induce the uptake of antimicrobials by disturbing or stressing the membrane, thus reducing the viability of microorganisms. However, its effectiveness depends on the time of exposure, type of treatment, food matrices, and type of microorganism. Indeed, Gram-positive and -negative bacteria have morphological differences: Gram-negative bacteria have a cell wall formed by a multi-layered structure: an outer membrane, lipopolysaccharide bilayer, and peptidoglycans [65]. On the other hand, Gram-positive bacteria have a single layer of peptidoglycan that is 20–80 nm thick [65,66].

Mechanical effects due to cavitation cause different types of physical damage to cell walls: Gram-negative bacteria are more sensible than positive ones. Indeed, microstreams’ action and shockwaves induce mass transfer processes and wall damage; hotspots cause local injury. Locally high temperatures can affect the integrity of layers [67,68]. As previously mentioned, the formation of -OH radicals intracellularly and H_2_O_2_ brings about the oxidation of intracellular amino acids (tyrosine, phenylalanine, tryptophan, histidine, methionine, and cysteine) and inhibition corresponding to specific functions [69]. Free radicals also cause chain reactions and consequently lipid oxidation. These reactions influence bacterial membrane fluidity, permeability, and deterioration; finally, when free radicals reach the intracellular space, they damage internal components and consequently the cell collapses [70]. In more detail, H_2_O_2_ and -OH attack the polysaccharide layer of the Gram-negative membrane wall, causing the scission of the glycoside backbone and the consequent fragmentation of the biopolymer and alteration of its function [71]. Nucleic acids can also be susceptible to oxidative stress by -OH; in fact, it can break the double helix or modify nitrogen bases [72] (see Figure 2).

Microbial inactivation can be influenced by different parameters, such as the nature of ultrasonic waves, food composition, temperature treatment, volume of food being processed, type of microorganism, and exposure time [73]. For these reasons, it is important to evaluate each individual microorganism with different parameters.

## 4. Pathogen *Escherichia coli*

In 2019, 7775 confirmed cases of Shiga toxin-producing *E. coli* (STEC) infections in humans were reported at the EU level by 27 EU countries [1]. The goal of ultrasound treatment against *E. coli* is wall damage, indicated as a morphological change during the treatment. Liu and coworkers studied the alterations of membrane permeability using an ultrasonic field [74]. They suggested that the outer membrane was the first target upon ultrasound treatment, and the inner membrane could be destabilized with an increase in time. Che et al. [73] evaluated the responses of bacterial cell membranes to ultrasound exposure with different parameters: 64, 191, 372, and 573 W/cm_2_, a frequency of 20 kHz, a pulsed mode of 2 sec: 2 sec. The outer membrane of *E. coli* presents robust and selective permeability [75]. Membrane fluidity, carrier transport, and membrane-bound enzymes are closely correlated with the integrity of the membrane [76]. In these bacteria, it is important to evaluate the absorbance of o-nitrophenol (ONP): ONP is hydrolyzed by β-galactosidase, which is an endoenzyme in *E. coli*, and progressive outward release from the cytoplasm occurs when the bacterial inner membrane is destroyed [77]. He et al. [78] evaluated morphological modifications through the usage of electron microscopy of *E. coli O157:H7 ATCC 35150* after ultrasound treatment with different times and different intensities. They noticed that morphological modification increases with the time of exposure. As illustrated in Table 2, the efficiency of ultrasound treatment in *E. coli* can be influenced by the treatment time, treatment power, and type of treatment (ultrasound alone or combined). It is very important to evaluate the food matrix. Indeed, every matrix could have a different response: liquid food was found to be more efficient than solid for inactivation treatment in *E. coli* [79,80].

## 5. *Salmonella* spp.

*Salmonella* spp. is the second most prevalent foodborne pathogen worldwide, as reported by the EFSA [1]. The applicability of ultrasound decontamination for *Salmonella* spp. has been an object of research since 1992, when Wrigley and Llorca evaluated the killing effect against *Salmonella* serovar Typhimurium ATCC 14028 by applying 35 and 40 kHz for 15 and 30 min in skim milk and liquid whole eggs [87]. They noticed also that liquid whole eggs protected *Salmonella* serovar Typhimurium from ultrasonic cavitation. Indeed, the food composition can influence ultrasound’s effects. Techathuvanan and D’Souza evaluated, through scanning electronic microscopy (SEM), the morphological differences between *Salmonella* spp. untreated and *Salmonella* spp. treated with high-intensity ultrasound after 5 and 30 min. Their work showed that there is a correlation between the time of exposure and the bacterial reduction after 1 min treatment for a pure culture [88].

The treatment efficiency of ultrasound alone or combined against *Salmonella* spp. could be variable. It could be influenced by the food matrix: the inactivation response of liquid food such as liquid whole eggs or rice beverages is more efficient than for solid foods as pork meat [85,89]. Extending the time of application leads to increased *Salmonella* spp. reduction [85,90] (Table 3).

## 6. *Listeria* spp.

The genus *Listeria* is naturally dispersed in soil, water, and manure [96]. The efficiency of ultrasound against *Listeria* spp. depends on the power, frequency, treatment time, temperature, and geometry reaction, and synergic effects with other technologies (essential oil, cold plasma, nanobubbles, etc.) [96]. Several studies show that the efficiency of the ultrasound inactivation of *Listeria* spp. is greater in liquid media, such as milk, broth, or juice [97,98,99,100,101]. Pan and coworkers [102] investigated the inactivation of *L. monocytogenes* by ultrasound and cold plasma; they studied the modification of the membrane fatty acid profile in correlation with different temperatures. They noticed a modification of the fluidity of the membrane and prevalence of fatty acids in relation to the different applications of treatments and temperatures and the presence of radical oxygen. Numerous studies evaluated the efficiency of ultrasound alone and combined with other technologies against *Listeria* spp. (see Table 4). It is important to observe that the same matrix, such as salmon treated by ultrasound and temperature, presented a different response in terms of microbial inactivation if it was raw or smoked. Ultrasound application showed greater effectiveness in ATCC (ATCC LM 19114, ATCC LM 15313, ATCC LM 19111, ATCC LM 7644, ATCC BAA 679, ATCC BAA 839, ATCC 13932, ATCC 19112) strains than wild ones (food origin) [103,104,105].

## 7. *Staphylococcus* spp.

*Staphylococcus* is normally present on animal and human skin and mucous membranes, and it could be ubiquitous in the environment. For this reason, it could pose an important risk for public health [108]. There is prolific scientific interest in ultrasound’s applicability against *Staphilococcus* spp., which was used as a study model for Gram-positive microorganisms to understand the modifications after treatment [109,110,111]. The effects of ultrasound technology, alone and combined, on *Staphylococcus aureus* are summarized in Table 5.

Mansyur et al. [111] analyzed the morphological differences in methicillin-resistant *S. aureus* (MRSA) applying low-power ultrasonic waves. They noticed that the power of the ultrasonic waves had a significant effect on the death percentage of MRSA (*p* = 0.0001), while the lethal power as found via regression was 8.432 watts. The death indicators of MRSA affected by ultrasonic waves were changes in shape (*p* = 0.005) and size (*p* = 0.70). Liao and coworkers [110] examined intracellular and extracellular changes in *S. aureus* (ATCC 25923): after ultrasound treatment, they evaluated the fluidity, integrity of the external membrane wall, intracellular and extracellular reactive oxygen species, and DNA damage. They explained that the major resistance of *S. aureus* against ultrasound could be explained by the thicker, more rigid, and robust properties of **Gram**-**positive** microbial cell envelopes [109,112]. The subpopulation of *S. aureus* lacking cell membrane integrity increased by 20.49% during 12 min of ultrasound treatment. Cell membrane potential is indispensable for normal energy transduction and nutrient uptake in microbial cells and is regarded as an important indicator of physiological activity [109]. The ultrasound treatment interferes with the lipid cell wall and consequently the bacterial growth process. The mechanical ultrasound power determines the separation of the multi-molecular complex and the stretching of the cell wall, limiting the elasticity; thus, the cell is torn, and bacteria die [111]. Few studies have been published on the ultrasound inactivation of *S. aureus*, an important foodborne pathogen associated with outbreaks worldwide. For this reason, this review article (Table 5) has underlined any substantial scientific criticisms about this important foodborne pathogen, which causes different infectious outbreaks in many geographical areas. This paper seeks to provide directions for further scientific investigations.

## 8. *Campylobacter* spp.

Campylobacteriosis is a zoonotic disease and humans could contract this illness via the consumption of raw poultry and water [116]. The ultrasound technology’s effects, alone and combined, on *Campylobacter* are summarized in Table 6.

Selwet [119] found *C. coli* in 21 out of 50 water samples. In his study, he evaluated sonication as an important tool for water decontamination. The research demonstrated that ultrasound application with a frequency of 80 kHz reduced the bacterial count from 6.86 log CFU/mL to 3.08 log CFU/mL, whereas a frequency of 37 kHz reduced the bacterial count from 6.75 log CFU/mL to 4.04 log CFU/mL. The study also underlined a temperature increase. Moazzami et al. [118] underlined that *Campylobacter* spp. reduction through ultrasound associated with steam is a good solution to avoid the risk of disease and preserve the quality of chicken. Manusavian and coworkers [122] valuated ultrasound’s effectiveness against *Campylobacter* spp. in 648 back, breast, and neck skin samples. The research noticed that there was a different reduction (*p* < 0.001) dependent on the sampling site (0.8, 1.1, and 0.7 log, respectively); it also evaluated samples after 8 days of refrigeration at 4 °C in control and steam-ultrasound-treated broilers to determine the contamination stability, and the results showed no changes in reductions during refrigeration, indicating that the reduced *Campylobacter* numbers remained stable in treated broilers.

## 9. *Vibrio* spp.

*Vibrio* species are major causes of fishery foodborne diseases worldwide, due to their presence in marine environments. Indeed, they are found in raw and ready-to-eat food products [123,124]. Thermal treatment is an effective method to reduce *Vibrio* spp. levels in seafood, but, as reported by Su et al. [125], the major disadvantage is the change in sensory characteristics. Considering the increasing consumer requests for fresh and nutritional food, and the consumption of shellfish such as oysters raw or cooked at a low level, non-thermal technology such as ultrasound could represent an important resource for bacterial inactivation [125]. In the literature consulted, all researchers used ultrasound alone or combined with slightly acidic electrolyzed water, temperature, and ozone (Table 7). As reported by Wang et al. [73], the major results are obtained through combined ultrasound and temperature (47 °C −204 W for 8 min), reducing the bacterial level by 4.01 log CFU/g.

Few studies have been published on the ultrasound inactivation of *Vibrio* spp., an important foodborne pathogen associated with outbreaks worldwide. For this reason, this review article (Table 7) has underlined any substantial scientific criticisms about this important foodborne pathogen, which causes different infectious outbreaks in many geographical areas. This paper aims to provide directions for further scientific investigations.

## 10. Pseudomonas spp.

Pseudomonas are found in water, soil, food, humans, plants, and surfaces, due to their versatility [129]. In food matrices, there is no health risk, but the presence of these pathogens causes an off-flavor due to producing volatile and amino acid metabolites and thermotolerant proteolytic enzymes that reduce the quality and shelf-life [130]. Because of the different characteristics of Pseudomonas spp., it is important to obtain an efficient technology for their inactivation. The application of ultrasound is not very efficient; in fact, it causes the insufficient reduction of bacteria as reported by Zhao et al. [131] (Table 8). On the other hand, its application combined with other technologies, such as temperature, is most efficient; in fact, the colony forming units decreased from 3 to 5 log CFU/gr ([132,133,134]) (Table 8). Greater exposure (expressed as time value) enhances the treatment efficiency, as reported by Kordowska-Wiater and Stasiak [82] (Table 8).

Ying and coworkers reported that combined treatment (ultrasound–temperature) against Pseudomonas fluorescens showed high efficacy for biofilm control: applying ultrasound (power > 80 W) and mild heat (up to 50 °C) caused the viable cell count to decrease. Indeed, ultrasound contributed to the release of biofilm bacteria in the environment and at the same time they exposed inner bacteria at the deep layer of the biofilm through shock waves, with acoustic streaming effects [135,136].

## 11. Conclusions

Ultrasound is an important technology to satisfy consumers’ requests and desire for “*fresh-like*”, safe, and healthy food. This technology preserves the nutritional, sensory, and compositional properties of food, and it is cheap and green. The environmental sustainability of this kind of physical food processing has attracted more attention among many industries. In more detail, many scientific studies, referring to these possibilities, highlighted a consistent reduction in carbon emissions in the atmosphere. This means a reduced impact on the so-called carbon footprint calculation.

Ultrasound is a noninvasive and cost-effective technique used to improve in terms of time other processes, such as cutting, cooking, freezing, drying, pickling/marinating, tenderization, and shelf-life. Ultrasound should represent a pivotal tool for foodborne pathogens’ (i.e., *Listeria monocytogenes*, *Salmonella* spp., *Staphylococcus* spp., *Vibrio* spp.) decontamination, with high potential due to its eco-friendly and non-thermal properties: the inactivation performance is variable with different microorganisms (bacteria, viruses, mycotoxins, and fungi) and food matrices. Nonetheless, research results suggest that parameters such as the frequency, intensity, treatment time, and treatment alone or combined with other technologies should be optimized for each food type. 

Several studies showed that the applicability of “*multiple hurdle technology*” is more effective than ultrasound alone; indeed, the combined action of two or more technologies is more efficient than the use of a single one [135,136,137].

Hence, future research must be directed towards the different inactivation mechanisms, microbial inactivation kinesis, and synergetic effects with other technologies.

## Figures and Tables

**Figure 1 foods-12-01212-f001:**
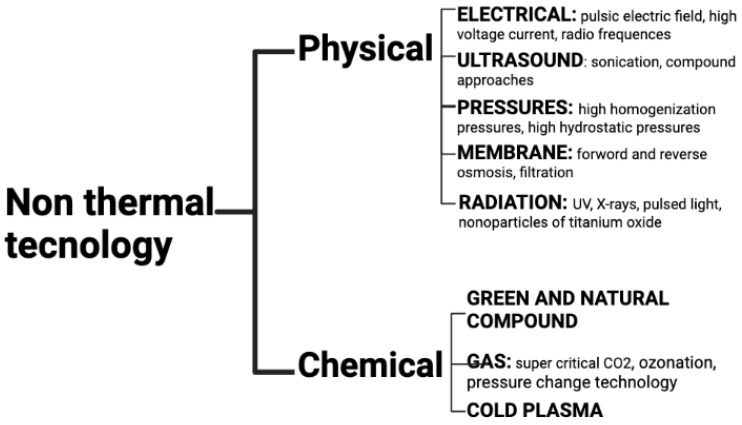
Overview of non-thermal technologies (created with Biorender.com).

**Figure 2 foods-12-01212-f002:**
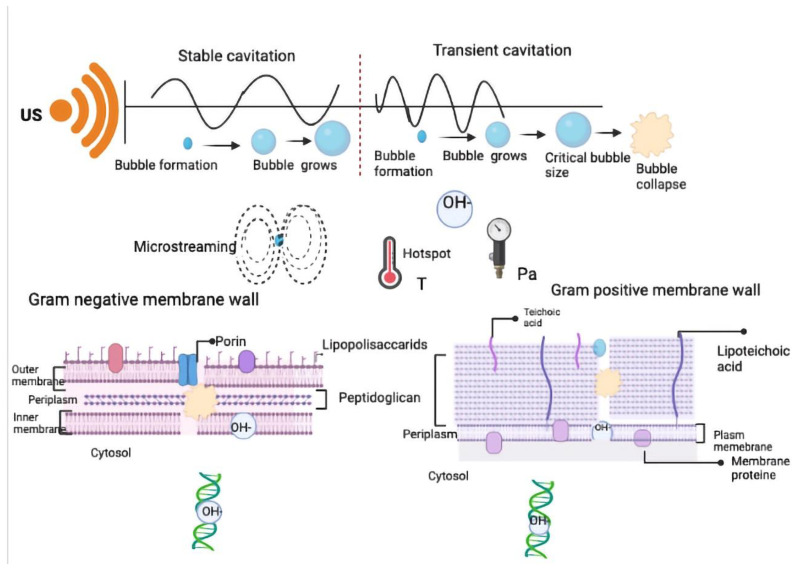
Mechanism of action of ultrasound against bacteria (created with Biorender.com).

**Table 1 foods-12-01212-t001:** Use of ultrasound in food industry: mechanisms, advantages, products.

Application	Conventional Method	Advantages	Ultrasound Principle	Products	Reference
Cutting	Knife	Small deformation Less cracks and crumbling	Cavitation phenomenon	Fragile and frozen foods Viscoelastic products Heterogenous products	[33] [34,35] [36]
Cooking	Stove Fried Water	Homogeneous cooking Less time	Uniform heat transfer	Poultry Beef Vegetables Fruit Crustaceans	[37,38] [39] [40] [39] [41]
Freezing	Ice	Less freezing time Homogeneous cooking Less damage to cells	Cavitation Fragmentation of large ice crystals Triggering secondary ice nucleation	Vegetables Meat Fish	[42] [39] [43]
Drying	Hot gas streaming Pulverization	Less time Improved heat transfer	Uniform heat transfer	Vegetables Meat Fish Fruit	[44] [45] [46] [47]
Pickling/marinating	Brine	Improved organoleptic quality Less time	Uniform heat transfer Microchannel	Meat Vegetables Cheese Fish	[48] [49] [50] [51]
Tenderization	Time	Improved meat tenderization	Acoustic cavitation	Meat Fish	[52] [53]

**Table 2 foods-12-01212-t002:** Ultrasound effects as single or combined non-thermal technology: *Escherichia coli*.

Organism	Matrix	Treatment	Parameter	Ultrasound Effects	Reference
*Escherichia coli* O157:H7	Fresh vegetables	Ultrasound at low frequency	P: 100 W, T: 7.0 min, and I: 50 w/cm_2_	Inactivation treatment (P: 100 W, T: 4 min, I: 10 w/cm_2_)	[80]
*Escherichia coli* O157:H7	Milk Orange juice	Ultrasound at low frequency Ultrasound + antimicrobial peptides	P: 40 W, 160 W, T 30 min, T60 min P: 40 W, 160 W, T 30, T60	Inactivation of inoculated *E. coli* Synergic effects	[81]
*Escherichia coli* O157:H7	Bacterial cell suspension	Ultrasound at low frequency	P: 0.667 and 6.67 W/mL, I: 25.5 and 255 W/cm_2_, T: 0, 5, 15, 25 min	Different time for low and high intensity	[77]
*Escherichia coli* O157:H7	Bacterial cell suspension	Ultrasound Ultrasound + nisin	P 20 W, 40 W, 60 W, and 80% by 20 kHz (total P: 950 W) 242.04 W, 484.08 W, 726.12 W, and 968.16 W/cm_2_; T: 15 min	Inactivation by ultrasound and with nisin	[82]
*Escherichia coli*	Cactus pear juice	Ultrasound at high frequency	P: 1500 W 20, 40, 60, and 80% by 20 kHz, t 2 sec 5 min	Inactivation 60%, 80% for 5 min	[83]
*Escherichia coli* (ATCC 11755)	Fresh carrot juice	Ultrasound + temperature	24 kHz, 120 μm, and 400 W with temperatures of 50, 54, and 58 °C and T: 0 to 10 min	5 log CFU/mL reduction after 2 min at 54 °C and 58 °C 3.5 log CFU/mL reduction after 10 min at 50 °C	[18]
*Escherichia coli* O157:H7	Beef	Ultrasound at low frequency	2.39, 6.23, 11.32, and 20.96 Wcm^−2^ 30, 60, 90, and 120 min	20.96 W cm^−2^ for 120 min was the optimal treatment for bacterial reductions	[84]
*Escherichia coli* K12 TEAG 1133	Pork meat	Ultrasound + NaCl	P: 95 W, T: 1 h	Treatment could assist current sodium reduction strategies, improving processing time and decontamination of brining tanks, increasing the shelf-life	[85]
*Escherichia coli*	Sliced shad *(Konosirus punctatus)*	Ultrasound Slightly acidic electrolyzed water + ultrasound	Ultrasound 37 kHz, 380 W 0, 50, and 100 min pH range 5.0–6.5, oxidation–reduction potential 650– 1000 mV, available chlorine concentration 10–80 mg/L containing 0, 15, and 30 ppm chlorine and ultrasound 37 kHz, 380 W for 0, 50, and 100 min	Treatment not sufficient 1.04–1.86 log CFU/g Reduction in T	[79]
*Escherichia coli* K12	Mackerel fillets	Ultrasound Ultrasound + plasma Ultrasound + peracetic acid Ultrasound + plasma-activated water + peracetic acid	25 kHz, 550 W, 10 min 25 kHz, 550 W, 10 min + 11 L/min 25 kHz, 550 W, 10 min + 200 ppm 25 kHz, 550 W, 10 min + 11 L/min, 10 min + 200 ppm	Inactivation of 0.38 CFU/g Inactivation of 0.2 CFU/g Inactivation of 0.59 CFU/g Inactivation of 0.59 CFU/g	[86]

**Table 3 foods-12-01212-t003:** Ultrasound effects as individual or combined non-thermal technology: *Salmonella* spp.

Organism	Matrix	Treatment	Parameter	Ultrasound Effects	Reference
*Salmonella* Typhimurium	Liquid whole egg	High-power ultrasound + lysozyme	35–45 °C and 605–968 W/cm_2_ for 5–35 min	Ultrasound and ultrasound + Lys caused a reduction of 3.31 and 4.26 log10 cycles	[89]
*Salmonella* Enteritidis	Liquid whole egg	High-power ultrasound	20 kHz HIU for 0, 1, 5, 10, and 30 min	Significant reduction in cells up to 3.6 log CFU/mL	[88]
*Salmonella* Enterica ATCC 35664	Rice beverage	Low-power ultrasound	20 kHz 130 W T 2, 6, 10 min P 40%, 60%, 100%	Confirmation of the strong effect of both power and time, although the correlation with the antibacterial action was not strictly linear	[91]
*Salmonella* spp.	Raw chicken meat	High-power ultrasound + carbon dioxide	40 kHz/30 min/40 °C	Inactivation of inoculated *Salmonella*	[92]
*Salmonella* Typhimurium	Chicken skin	Ultrasound + ethanol	Ethanol 70% + ultrasound (37 kHz, 380 W)	Inactivation of inoculated *Salmonella*, change in Hunter color and skin texture	[93]
*Salmonella* Typhimurium CICC2295	Pork meat	Ultrasound	20 kHzf T: 10, 20, 30 min	1–4.3 and 1–4.6 log CFU/g reduction	[85]
*Salmonella* Typhimurium	Chicken skin	Ultrasound Ultrasound + peroxyacetic acid	37 kHz, 380 W 5 min 37 kHz, 380 W 5 min + 50–200 ppm	Treatment not sufficient Reductions of 2.21 and 2.08 log CFU/g	[94]
*Salmonella* Typhimurium ATCC 14028	Low-fat and high-fat milk	Ultrasound Ultrasound + cinnamon essential oil	24 kHz and 400 W power at 124 μm (100%) wave amplitude 15 min 24 kHz and 400 W power at 124 μm (100%) wave amplitude 15 min + cinnamon	Reduction of 1.6 log cycle Reduction of 2.7 log CFU/mL in low-fat milk and 3.8 log CFU/mL in high-fat milk	[95]
*Salmonella* Enterica Anatum	Chicken skin	Ultrasound Ultrasound + lactic acid aqueous solution	40 kHz, 2.5 W/cm_2_ for 3 or 6 min 40 kHz, 2.5 W/cm_2_ for 3 or 6 min	0.6 log CFU/cm_2_ 1 log CFU/cm_2_ 1.6 log CFU/cm_2_ 2.7 log CFU/cm_2_	[90]

**Table 4 foods-12-01212-t004:** Ultrasound effects as single or combined non-thermal technology: *Listeria* spp.

Organism	Matrix	Treatment	Parameter	Ultrasound Effects	Reference
*Listeria innocua*	Blueberry	Ultrasound + carvacrol + carbonated water	20 kHz 500 W, 1/3.3 MHz 10 W + solution of carvacrol	After 10 min of treatment with 2 mM carvacrol (CR), carbonated water (CW), 20 kHz ultrasound (20 kHz), or 1 MHz ultrasound (1 MHz) alone, there was a 2.4–2.6 log CFU/g reduction (*P* < 0.05) in bacteria from blueberry surface from the initial load of 5.2 log CFU/g	[106]
*Listeria* *monocytogenes* LM ATCC 19114, LM ATCC 15313, LM ATCC 19111, LM ATCC 7644	Smoked salmon	Ultrasound + temperature	20 kHz, 100% amplitude, 20 °C, 25 °C, 30 °C, 40 °C, 50 °C, T: 5, 10, 15 min	Inactivation was 2.02, 2.12, and 2.44 log CFU/g at 30 °C for 15 min, at 40 °C for 15 min, and at 50 °C for 5 min	[104]
*L. monocytogenes* ATCC19115	Bacterial cell suspension	Ultrasound + cold plasma + temperature	500 W and 40 kHz T 0, 2, 5, 10 min + plasma treatment 2 min	Inactivation by ultrasound and cold plasma, increasing temperature	[102]
*Listeria innocua*	Mackerel fillets	Ultrasound Ultrasound + plasma Ultrasound + peracetic acid Ultrasound + plasma-activated water + peracetic acid	25 kHz, 550 W, 10 min 25 kHz, 550 W, 10 min + 11 L/min 25 kHz, 550 W, 10 min + 200 ppm 25 kHz, 550 W, 10 min + 11 L/min, 10 min + 200 ppm	Inactivation of 0.33 CFU/g Inactivation of 0.20 CFU/g Inactivation of 0.72 CFU/g Inactivation of 0.65 CFU/g	[86]
*Listeria* *monocytogenes* ATCC 19115	Low-fat and high-fat milk	Ultrasound Ultrasound + cinnamon essential oil	24 kHz and 400 W power at 124 μm (100%) wave amplitude 15 min 24 kHz and 400 W power at 124 μm (100%) wave amplitude 15 min + cinnamon	Reduction of 2.5 and 3 log cycles Reduction of 4.3 and 4.5 log cycles	[95]
*Listeria innocua*	Spinach leaves	Ultrasound Ultrasound + nanobubble		Did not significantly reduce bacteria More than 6 log CFU/mL reduction after 15 min	[107]
*Listeria* *monocytogenes*	Salmon filets	Ultrasound	200 W, 45 kHz	Reduction of 0.6 log CFU/mL	[103]

**Table 5 foods-12-01212-t005:** Ultrasound effects as single or combined non-thermal technology: *Staphylococcus aureus*.

Organism	Matrix	Treatment	Parameter	Ultrasound Effects	Reference
*Staphylococcus aureus* ATCC 25923	Broth colony	Ultrasound	30 kHz 100 W from 5 to 3° min	Not sufficient	[112]
*Methicillin-resistant* *Staphylococcus aureus*	Broth colony	Ultrasound	20 kHz 2, 3, 4, 5, or 6 watts for 2 min	Lethal power = 8.432 watts	[113]
*Staphylococcus aureus* ATCC 25923	Broth colony	Ultrasound	198 W, 252 W/cm^2^, 20 kHz	Bacterial damage	[110]
*Staphylococcus aureus*	Chicken breast	Ultrasound	Ultrasonic bath 9.6 W/cm^2^ /40 kHz/0, 30, and 50 min/5 °C	*S. aureus* increased	[114]
*Staphylococcus aureus*	Milk	Ultrasound + temperature	20 kHz, 600 W, 120 lm, 12 min + 60 C	0.94 log CFU ml1	[115]

**Table 6 foods-12-01212-t006:** Ultrasound effects as single or combined non-thermal technology: *Campylobacter* spp.

Organism	Matrix	Treatment	Parameter	Ultrasound Effects	Reference
*Campylobacter jejuni*	Chicken carcasses	Ultrasound + vacuum	1200 W/130 Hz/15 min + 0.1% cetylpyridinium chloride or 0.01% sodium hypochlorite and a vacuum of −0.02 MPa	From 0.94 to 1.19 log10 MPN (most probable number)/10 gr	[117]
*Campylobacter jejuni*	Chicken carcasses	Ultrasound + steam	30 to 40 kHz and steam at 84 to 85 °C	0.5–0.8 log CFU/g	[118]
*Campylobacter jejuni*	Chicken skin	Ultrasound Ultrasound + peroxyacetic acid	37 kHz, 380 W 5 min 37 kHz, 380 W 5 min + 50–200 ppm	Not sufficient at 0.25 log CFU/g Reduction of 2.08 log CFU/g	[94]
*Campylobacter coli* *ATCC 33559*	Water	Ultrasound	37 kHz and 80 kH + 5 min	Frequency of 80 kHz reduction from 6.86 log CFU/mL to 3.08 log CFU/mL, 37 kHz reduction 6.75 log CFU/mL to 4.04 log CFU/mL	[119]
*Campylobacter jejuni*	Raw chicken	Ultrasound + temperature	4, 25, and 54 °C, 40 kHz, ultrasound power of 120 W, 1, 2 or 3 min	Reduction	[120]
*Campylobacter jejuni*	Chicken carcass	Ultrasound + temperature	90–94 °C and + t 30–40 kHz	Reduction of 0.7 log_10_ CFU	[121]

**Table 7 foods-12-01212-t007:** Ultrasound effects as single or combined non-thermal technology: *Vibrio* spp.

Organism	Matrix	Treatment	Parameter	Ultrasound Effects	Reference
*Vibrio* *paraemoliticus* KCTC 2471	Sliced shad *(Konosirus punctatus)*	Ultrasound Slightly acidic electrolyzed water + ultrasound	Ultrasound 37 kHz, 380 W 0, 50, and 100 min pH range 5.0–6.5, oxidation–reduction potential 650– 1000 mV, available chlorine concentration 10–80 mg/L containing 0, 15, and 30 ppm chlorine and ultrasound 0.37 kHz, 380 W, 0, 50, and 100 min	Treatment not sufficient 1.02–1.42 log CFU/g reduction	[126]
*Vibrio* *paraemoliticus* ATCC 33847	Raw peeled shrimp	Ultrasound Ultrasound + temperature	0, 96, 150, and 204 W 0, 96, 150, and 204 W, 47, 50, and 53 °C	Limited reduction of 0.59, 0.60, and 0.68 log CFU/g 47 °C reductions of 1.76, 2.63, and 4.01 log CFU/g 96, 150, and 204 W, respectively, for 8 min	[82]
*Vibrio* *vulnificus*	Oysters (*Crossostrea virginica)*	Ultrasound	100 W, 500 W/cm for 30 min	Treatment not sufficient	[127]
*Vibrio cholerae*	Broth solution	Ultrasound Ultrasound + ozone	40 kHz, 150 W 10 min	Treatment not sufficient Treatment not sufficient	[128]

**Table 8 foods-12-01212-t008:** Ultrasound effects as single or combined non-thermal technology: Pseudomonas spp.

Organism	Matrix	Treatment	Parameter	Ultrasound Effects	Reference
*Pseudomonas* *fluorescens*	Mackerel fillets	Ultrasound Ultrasound + plasma Ultrasound + peracetic acid Ultrasound + plasma-activated water + peracetic acid	25 kHz, 550 W, 10 min 25 kHz, 550 W, 10 min + 11 L/min 25 kHz, 550 W, 10 min + 200 ppm 25 kHz, 550 W, 10 min + 11 L/min, 10 min + 200 ppm	Inactivation of 0.50 CFU/g Inactivation of 0.13 CFU/g Inactivation of 0.46 CFU/g Inactivation of 0.30 CFU/g	[86]
*Pseudomonas* *fluorescens*	Chicken skin	Ultrasound Ultrasound + lactic acid aqueous solution	40 kHz, 2.5 W/cm^2^ for 3 or 6 min 40 kHz, 2.5 W/cm^2^ for 3 or 6 min	0.5 log CFU/cm^2^ 1 log CFU/cm^2^ 3 log CFU/cm^2^ 4.1 log CFU/cm^2^	[94]
*Pseudomonas* *fluorescens*	Milk	Ultrasound + temperature	20 kHz, 150 W + 62 °C	3.1 CFU/mL	[131]
*Pseudomonas* *fluorescens*	Raw milk	Ultrasound + temperature	60 kV/cm, 200 μs, 40 °C (Tin)	5 CFU/mL	[132]
*Pseudomonas* *putida*	Milk	Ultrasound	20 kHz, 100 W	Bacteriostatic effect	[133]
*Pseudomonas fluorescens*	Fresh-cut kale	Ultrasound + temperature	100 W/L at 25, 40, or 50 °C	Reduced by 3 log CFU/mL	[134]

## Data Availability

Not applicable.

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
