# Peer review of "Ultrasound Technology as Inactivation Method for Foodborne Pathogens: A Review"

_foods, 2023, doi:10.3390/foods12061212_

Round 1

Reviewer 1 Report

The Review (Manuscript ID: foods-2256272) by Lautery et al., provided an interesting and comprehensive evaluation of the use of ultrasound as an effective methodology for inactivating foodborne bacterial pathogens.  The review provided a general description of the methodology followed by detailed narratives focusing on a particular species on the effect of ultrasound for inactivating these bacterial pathogens. 

Other comments:

For this Reviewer, is there a reason for abbreviating ultrasound as “US”? This Reviewer cannot see the added value of abbreviating the word, and I find it to be necessary. Also, the abbreviation was not used consistently throughout the manuscript. As few examples, lines 52, 66 or 71 or Table 3 didn’t use the abbreviation “US” but instead “ultrasound”.  I strongly suggest removing the abbreviation “US” in the entire manuscript and just writing the word “ultrasound”. Too many abbreviations make the text difficult to read!!

Line 57:  Suggest adding “the” after “In” for “In food industry”.

Line 74: Change to plural “types”.

Line 83: What is “metal horns”? Please elaborate this term to enable readers from different backgrounds to understand the statement.

Line 88: Correct typographical error “uses ais as”.

Lines 96-97, and throughout the entire manuscript text: For Gram positive or negative, Gram should be in lowercase letter. Please remove the hyphen in line 97 for gram-negative.

Lines 122-123: Suggest adding “each” after “singularly”.

Lines 124-145: Given that not all Escherichia coli are pathogenic, I suggest specifying the pathogen, Shiga toxin-producing E. coli, which is examined in this section of the Review.  Italicize “E.coli in line 125. Change the title of this sub-section to the pathogen not just Escherichia coli.  In lines 138-144 on the research by He et al., was it STEC the pathogen examined? Please revise text.

Line 143: Add “to” after “important”.

Line 153: Correct misspelled word “Tiphimurium”.

Line 183: Need some clarification on the statement “effectiveness in ATCC strains than wild ones”.  By using the term “ATCC” is it meant reference strains? What are “wild” strains? Is it meant by strains recovered from wildlife or environmental sources? Please elaborate on the findings by MiksÌŒ-Krajnik et al, and Pennisi, et al (refs 65 and 96).

Line 190: What is “this” on “this is ubiquitous” and what is “it” on “it could be”? Is the statement referring to Staphylococcus?

Lines 216-220 and 256-260:  Suggest deleting the paragraph or just change it to say something like “limited studies have been published on the inactivation of S. aureus (or Vibrio for lines 256-260), an important foodborne pathogen associated with outbreaks worldwide.”

Line 222: Correct the misspelled word “Campilobacter”.

Line 223: Insert “and” after “disease,”.  Insert “consumption” after “by”.

Lines 229-242: Avoid using third-person pronouns (he or his) in a manuscript. State the findings as “the study” or “The research demonstrated”…

Line 264: Correct heading.

Line 265: Correct beginning of sentence since it is starting with a period.

Line 265: Remove “they are isolates” and replace “are found”. Remove colon after ubiquitous.

Lines 266-269: Please rewrite the sentence since it is not grammatically correct and difficult to read. Please add a period after “animal”.  Suggest stating “In food matrices, there is no health risk but the presence of these pathogens cause….”.

Line 286: Please edit the sentence since ultrasound is not an instrument.  As described at the beginning of the manuscript, ultrasound is a technique or a intervention strategy.

Line 297: Replace “A lot of” with “Several studies”.  Replace “is more effective” for “as more effective”.

Line 298: Change to “two or more technologies are…”

Lines 299-303: The statements need to be edited to improve the sentence construction. Please revise since there are run-on sentences (two independent clauses put together without proper punctuation or appropriate conjunction).

Tables 2-8 Column entitled “Treatment”: Given that the entire manuscript is about ultrasound, the column “Treatment” should be labeled as “Ultrasound Treatment” to avoid having in each column “US”. Then highlight in each scenario whether the treatment was either low power, high power, or by adding an additional compound (e.g., essential oil added).

Author Response

Reviewer 1

All authors want to express their appreciation to considerer the manuscript entitled “Ultrasound technology as inactivation method of foodborne pathogens: A review” for publication in the FOODS Journal in the Food Microbiology Section.

All reviewers’ suggestions resulted interesting and important to improve manuscript’s scientific quality.

In the following paragraphs we reported all changings.

Reviewer 1 suggestion

For this Reviewer, is there a reason for abbreviating ultrasound as “US”? This Reviewer cannot see the added value of abbreviating the word, and I find it to be necessary. Also, the abbreviation was not used consistently throughout the manuscript. As few examples, lines 52, 66 or 71 or Table 3 didn’t use the abbreviation “US” but instead “ultrasound”.  I strongly suggest removing the abbreviation “US” in the entire manuscript and just writing the word “ultrasound”. Too many abbreviations make the text difficult to read!!

Authors (lines 11, 15, 46, 50, 59, 72, 75, 87, 92, 128, 133, 141, 143, 145, 149…): All abbreviations “US” have been replaced with “ultrasounds”.

Line 57:  Suggest adding “the” after “In” for “In food industry”.

Authors (line 57): As requested, “the” was added.

Line 74: Change to plural “types”.

Authors (line 74): As requested, we changed.

Line 83: What is “metal horns”? Please elaborate this term to enable readers from different backgrounds to understand the statement.

Authors (lines 83-84): We explain “metal horns” (Corrotto and Cintas, 2007, The Combined Use of Microwaves and Ultrasound: Improved Tools in Process Chemistry and Organic Synthesis).

Line 88: Correct typographical error “uses ais as”.

Authors (line 89): Corrected.

Lines 96-97, and throughout the entire manuscript text: For Gram positive or negative, Gram should be in lowercase letter. Please remove the hyphen in line 97 for gram-negative.

Authors (lines 96-97; 98; 100; 102; 117): Corrected.

Lines 122-123: Suggest adding “each” after “singularly”.

Authors (lines 128-129): “each” has been added.

Lines 124-145: Given that not all Escherichia coli are pathogenic, I suggest specifying the pathogen, Shiga toxin-producing E. coli, which is examined in this section of the Review.  Italicize “E. coli in line 125. Change the title of this sub-section to the pathogen not just Escherichia coli.  In lines 138-144 on the research by He et al., was it STEC the pathogen examined? Please revise text.

Authors (lines 130-152): Subsection title has been changed; all E. coli strains have been written in the italics form. Supplementary information about He et al. has been added.

Line 143: Add “to” after “important”.

Authors (line 151): “to” has been added.

Line 153: Correct misspelled word “Tiphimurium”.

Authors (line 160): “Tiphimurium” has been corrected.

Line 183: Need some clarification on the statement “effectiveness in ATCC strains than wild ones”.  By using the term “ATCC” is it meant reference strains? What are “wild” strains? Is it meant by strains recovered from wildlife or environmental sources? Please elaborate on the findings by MiksÌŒ-Krajnik et al, and Pennisi, et al (refs 65 and 96).

Authors (lines 200-202): We added reference ATCC and wild strains.

Line 190: What is “this” on “this is ubiquitous” and what is “it” on “it could be”? Is the statement referring to Staphylococcus?

Authors (lines 208-209): Sentence was explained.

Lines 216-220 and 256-260:  Suggest deleting the paragraph or just change it to say something like “limited studies have been published on the inactivation of S. aureus (or Vibrio for lines 256-260), an important foodborne pathogen associated with outbreaks worldwide.”

Authors (lines 269-271): modified.

Line 222: Correct the misspelled word “Campilobacter”.

Authors (line 275): Corrected.

Line 223: Insert “and” after “disease,”.  Insert “consumption” after “by”.

Authors (line 276): corrected.

Lines 229-242: Avoid using third-person pronouns (he or his) in a manuscript. State the findings as “the study” or “The research demonstrated”…

Authors (lines 356-369): This part has been changed.

Line 264: Correct heading.

Authors (line 427): corrected.

Line 265: Correct beginning of sentence since it is starting with a period.

Authors (428-430): Corrected.

Line 265: Remove “they are isolates” and replace “are found”. Remove colon after ubiquitous.

Authors (line 428): Replaced.

Lines 266-269: Please rewrite the sentence since it is not grammatically correct and difficult to read. Please add a period after “animal”.  Suggest stating “In food matrices, there is no health risk but the presence of these pathogens cause….”.

Authors (lines 429-430): Modified.

Line 286: Please edit the sentence since ultrasound is not an instrument.  As described at the beginning of the manuscript, ultrasound is a technique or a intervention strategy.

Authors (line 507): The sentence has been edited.

Line 297: Replace “A lot of” with “Several studies”.  Replace “is more effective” for “as more effective”.

Authors (line 530): Replaced.

Line 298: Change to “two or more technologies are…”

Authors (line 531): Changed.

Lines 299-303: The statements need to be edited to improve the sentence construction. Please revise since there are run-on sentences (two independent clauses put together without proper punctuation or appropriate conjunction).

Authors (line 533): Edited.

Tables 2-8 Column entitled “Treatment”: Given that the entire manuscript is about ultrasound, the column “Treatment” should be labeled as “Ultrasound Treatment” to avoid having in each column “US”. Then highlight in each scenario whether the treatment was either low power, high power, or by adding an additional compound (e.g., essential oil added).

Authors: Modified.

We confirm that neither the manuscript nor any parts of its content are currently under consideration or published in another journal.

All authors have approved the manuscript and agree with its submission to the FOODS Journal.

We appreciate the possibility to publish our paper and believe that our manuscript will be of interest to You and to the readers of Your journal.

Thank You for Your time and attention.

Best regards,

Gianluigi Ferri

Doctor in Veterinary Medicine (D.V.M.)

Ph.D. Student in Food Inspection

Department of Veterinary Medicine; University of Teramo, Italy.

Reviewer 2 Report

The manuscript is describing the effects of ultrasound on food pathogens, providing some very useful technical info for the food industry. The references are recent and suitable, the figures are clear and informative and the methodology is sound.

Few minor corrections:

-Line 12: cell wall damage

- In Figure 1, correct 'ultrasuond' to 'ultrasound'

- In Figure 1, define NNT

- In Figure 1, move the flow chart to the centre of the box. 

- Line 49: mechanisms of action

- Line 91: microorganisms

-Line 102: Gram-negative

-In chemical formulae, use subscript for numbers e.g. in CO2, H2O2

-Figure 2: define US

Figure 2: 'hotspot' and 'bubble collapse' should start with upper case for consistency

-Line 297, A lot of studies

-Sections for E. coli and Salmonella are both numbered '4'. Please correct.

-Line 293: Your review has not elaborated on fungi, mycotoxins and viruses, please provide some references and more details in the text

-Conclusion

Author Response

Reviewer 2

All authors want to express their appreciation to considerer the manuscript entitled “Ultrasound technology as inactivation method of foodborne pathogens: A review” for publication in the FOODS Journal in the Food Microbiology Section.

All reviewers’ suggestions resulted interesting and important to improve manuscript’s scientific quality.

In the following paragraphs we reported all changings.

Reviewer 2 suggestions

The manuscript is describing the effects of ultrasound on food pathogens, providing some very useful technical info for the food industry. The references are recent and suitable, the figures are clear and informative and the methodology is sound.

Line 12: cell wall damage.

Authors (line 11): “Cell” has been added.

In Figure 1, correct 'ultrasuond' to 'ultrasound'

Authors: Figure 1 corrected.

In Figure 1, define NNT.

Authors: Defined.

In Figure 1, move the flow chart to the centre of the box.

Authors: Modified.

Line 49: mechanisms of action.

Authors (line 49): added.

Line 91: microorganisms

Authors (line 91): modified.

Line 102: Gram-negative

Authors: modified.

In chemical formulae, use subscript for numbers e.g. in CO2, H2O2.

Authors: modified.

Figure 2: define US

Authors: defined.

Figure 2: 'hotspot' and 'bubble collapse' should start with upper case for consistency

Authors: modified.

Line 297, A lot of studies.

Authors (line 507): modified.

Sections for E. coli and Salmonella are both numbered '4'. Please correct.

Authors (line 169): Corrected.

Line 293: Your review has not elaborated on fungi, mycotoxins and viruses, please provide some references and more details in the text

Authors: The present review article focused exclusively on ultrasound effects on bacterial strains.

Conclusion

Authors: Conclusion section has been changed.

We confirm that neither the manuscript nor any parts of its content are currently under consideration or published in another journal.

All authors have approved the manuscript and agree with its submission to the FOODS Journal.

We appreciate the possibility to publish our paper and believe that our manuscript will be of interest to You and to the readers of Your journal.

Thank You for Your time and attention.

Best regards,

Gianluigi Ferri

Doctor in Veterinary Medicine (D.V.M.)

Ph.D. Student in Food Inspection

Department of Veterinary Medicine; University of Teramo, Italy.
